# Mammaglobin-A Expression Is Highly Specific for Tumors Derived from the Breast, the Female Genital Tract, and the Salivary Gland

**DOI:** 10.3390/diagnostics13061202

**Published:** 2023-03-22

**Authors:** Natalia Gorbokon, Patrick Timm, David Dum, Anne Menz, Franziska Büscheck, Cosima Völkel, Andrea Hinsch, Maximilian Lennartz, Andreas M Luebke, Claudia Hube-Magg, Christoph Fraune, Till Krech, Patrick Lebok, Till S Clauditz, Frank Jacobsen, Guido Sauter, Ria Uhlig, Stefan Steurer, Sarah Minner, Andreas H. Marx, Ronald Simon, Eike Burandt, Christian Bernreuther, Doris Höflmayer

**Affiliations:** 1Institute of Pathology, University Medical Center Hamburg-Eppendorf, D-20246 Hamburg, Germany; 2Institute of Pathology, Clinical Center Osnabrueck, D-49076 Osnabrueck, Germany; 3Institute of Pathology, Academic Hospital Fuerth, D-90766 Fuerth, Germany

**Keywords:** mammaglobin-A, immunohistochemistry, tissue microarray, diagnostic marker, human cancers

## Abstract

Human mammaglobin-A (SCGB2A2) is a secretory protein with an unknown function that is used as a diagnostic marker for breast cancer. However, other tumors can also express mammaglobin-A. To comprehensively study patterns of mammaglobin-A expression, a tissue microarray containing 16,328 samples from 128 different tumor types as well as 608 samples of 76 different normal tissue types was analyzed using immunohistochemistry. Mammaglobin-A positivity was found in only a few normal tissues, including luminal cells of the breast as well as endocervical and endometrial glands. In tumor tissues, 37 of 128 tumor categories showed mamma-globin-A staining, 32 of which were derived from one of four organs: breast (6 tumor categories), endometrium (5 tumor categories), ovary (5 tumor categories), and salivary glands (16 tumor categories). Only five additional tumor types showed occasional weak mammaglobin positivity, including medullary thyroid cancer, teratoma of the testis, squamous cell carcinoma of the skin and pharynx, and prostatic adenocarcinoma. Among 1139 evaluable invasive breast carcinomas of no special type, low mammaglobin-A immunostaining was linked to high BRE grade (*p* = 0.0011), loss of estrogen and progesterone receptor expression (*p* < 0.0001 each), and triple-negative status (*p* < 0.0001) but not to patient survival. In endometrial cancer, mammaglobin-A loss was linked to an advanced tumor stage (*p* = 0.0198). Our data characterize mammaglobin-A as a highly specific marker for tumors derived from either the breast, female genitals, or salivary gland.

## 1. Introduction

Human mammaglobin-A was first described in 1998 as one of several proteins that were differentially expressed between breast cancers and matched normal breast tissues [1]. Together with more than 20 related proteins, termed secretoglobins, mammaglobin-A belongs to the uteroglobin/Clara cell protein family [2,3,4]. The function of secretoglobins is not well understood. They seem to be involved in cell signaling, immune response, and chemotaxis and may also serve as transporters for steroid hormones in humans [5,6,7,8,9,10,11]. The latter may particularly apply to mammaglobin-A as it is capable of binding steroid-like molecules [12,13]. Mammaglobin-A is encoded by the secretoglobin family 2A member 2 (SCGB2A2) gene located on chromosome 11q12 and translates into a protein of 93 amino acids. In normal tissues, mammaglobin-A is expressed almost exclusively by the mammary gland [1]. Due to its frequent upregulation in breast cancers [14], mammaglobin-A has been proposed as a promising target for therapy in these tumors [15]. The impact of mammaglobin overexpression on breast cancer aggressiveness is disputed, with studies reporting a tumor promoting effect [16], tumor suppressive effect [17], or no effect [18].

Due to its preferential expression in breast epithelial cells, mammaglobin-A immunohistochemistry has become an established diagnostic tool for recognizing metastatic breast cancer tissue [19,20]. However, the accumulated data on the prevalence of mammaglobin-A expression in tumors are controversial. For breast cancer, reported mammaglobin-A positivity rates range from 59 to 100% in lobular breast carcinomas [21,22] and from 25 to 94% in invasive breast carcinomas of no special type [23,24]. Immunohistochemical mammaglobin-A positivity has also been found in 11–76% of endometrium carcinomas of the uterus [25,26], 37–100% of ovarian carcinomas [26,27], and in 0–36% of prostatic adenocarcinomas [14,28]. These conflicting data are likely due to the use of different antibodies, immunostaining protocols, and criteria for the categorization of mammaglobin-A staining results in these studies.

To better understand the prevalence and significance of mammaglobin-A expression in cancer, a comprehensive study analyzing large numbers of neoplastic and non-neoplastic tissues under highly standardized conditions is needed. Therefore, we studied mammaglobin-A expression in more than 16,000 tumor tissue samples from 128 different tumor types and subtypes as well as 76 non-neoplastic tissue categories using immunohistochemistry (IHC) in a tissue microarray (TMA) format.

## 2. Materials and Methods

### 2.1. Tissue Microarrays (TMAs)

Tissue microarrays composed of normal and tumorous tissues were employed for this study. The normal tissue TMA contained 8 samples from 8 different donors from each of 76 different normal tissue types. The tumor TMAs contained a total of 16,328 primary tumors from 128 tumor types and subtypes. Histopathological data, including grade and pT and pN status, were available from 524 ovarian cancers, 259 endometrium cancers, and 1475 breast cancers. The breast cancer dataset also included molecular information on ER, PR, and HER2 as well as follow-up information on a subset of 877 patients with a median follow-up time of 49 months (range 1–88). The composition of both normal and tumor TMAs is described in Section 3. All samples were from the archives of the Institutes of Pathology, University Hospital of Hamburg, Germany; the Institute of Pathology, Clinical Center Osnabrueck, Germany; and Department of Pathology, Academic Hospital Fuerth, Germany. Tissues were fixed in 4% buffered formalin and then embedded in paraffin. The TMA manufacturing process was described previously in detail [29,30]. A single tissue core measuring 0.6 mm diameter per tumor was used to manufacture the TMA. The use of archived remnants of diagnostic tissues for manufacturing of TMAs and their analysis for research purposes as well as patient data analysis was approved by local laws (HmbKHG, §12) and the local ethics committee (Ethics Commission Hamburg, WF-049/09). All work was carried out in compliance with the Helsinki Declaration.

### 2.2. Immunohistochemistry (IHC)

Freshly cut TMA sections were immunostained on one day and in one experiment. Two different primary antibodies were used for mammaglobin-A detection in normal tissues: MSVA-457R (rabbit recombinant, MS Validated Antibodies, Hamburg, Germany, #2668-457R-01) and clone 305-1A5 (mouse monoclonal, FLEX RTU; Agilent, Santa Clara, CA, USA, #GA074). For tumor tissue analysis, only one antibody (MSVA-475R) was used. For MSVA-457R, staining was performed manually. Slides were deparaffinized and exposed to heat-induced antigen retrieval for 5 min in an autoclave at 121 °C in pH 9 buffer. Primary antibody was applied at 37 °C for 60 min at a dilution of 1:150. Bound antibody was then visualized using the EnVision Flex kit (Agilent, Santa Clara, CA, USA, #52023) according to the manufacturer’s directions. For 305-1A5 (RTU), slides were stained in a DAKO Autostainer Link 48 after Flex-high (pH 9) antigen retrieval (#GV804) using a protocol recommended by Dako/Agilent. For tumor tissues, the percentage of positive neoplastic cells was estimated, and the staining intensity was semi-quantitatively recorded (0, 1+, 2+, and 3+). For statistical analyses, the staining results were categorized into four groups as described before [31]. Tumors without any staining were considered negative. Tumors with 1+ staining intensity in ≤70% of tumor cells or 2+ intensity in ≤30% of tumor cells were considered weakly positive. Tumors with 1+ staining intensity in >70% of tumor cells, 2+ intensity in 31–70%, or 3+ intensity in ≤30% of tumor cells were considered moderately positive. Tumors with 2+ intensity in >70% or 3+ intensity in >30% of tumor cells were considered strongly positive.

### 2.3. Statistics

Statistical calculations were performed using JMP 14 software (SAS Institute Inc., Cary, NC, USA). Contingency tables and the chi² test were performed to search for associations between mammaglobin-A and tumor phenotype. Survival curves were calculated according to Kaplan–Meier. The log-rank test was applied to detect significant differences between groups. A *p* value of ≤0.05 was considered statistically significant.

## 3. Results

### 3.1. Technical Issues

A total of 14,232 (87.2%) of 16,328 tumor samples were interpretable in our TMA analysis. The remaining 2096 (12.8%) samples were not interpretable due to the lack of unequivocal tumor cells or a lack of the entire tissue spot. On the normal tissue TMA, enough samples (≥4) were always analyzable per tissue type to determine mammaglobin-A staining patterns.

### 3.2. Mammaglobin-A in Normal Tissue

The normal tissues were analyzed using both antibodies. Using the antibody MSVA-457R, mammaglobin-A immunostaining was only seen in a few normal tissue types, including luminal cells of the breast (not in all cells and with variable intensity), endocervical glands (mostly intense but not all glands in all patients), endometrium (not in all cells and with variable intensity), scattered epithelial cells in the fallopian tube (moderate staining intensity), eccrine glands of the skin (weak to moderate), a few scattered cells in the salivary glands (weak to moderate), and in some principal and clear cells of the epididymis (weak to moderate). Mammaglobin staining was found to be most intense in endocervical and endometrial glands, where the staining often also involved stroma components. Such stroma staining may reflect a diffusion/contamination artifact caused by a “spill-over” of the highly abundant mammaglobin-A protein into adjacent structures, which is potentially facilitated by pre-fixation tissue damage. Examples of mammaglobin-A-positive normal tissues are shown in Figure 1. Mammaglobin-A staining was completely lacking in the muscle, myometrium, ovary, fat, non-keratinizing squamous epithelium of the lip, oral cavity, tonsil, ectocervix, esophagus, urothelium, transitional epithelium of the anal canal, decidua, placenta, lymph node, spleen, thymus, stomach, duodenum, ileum, appendix, colon, rectum, gall bladder, liver, Brunner gland, bronchial gland, kidney, prostate, seminal vesicle, testis, respiratory epithelium, lung, adrenal gland, parathyroid gland, brain, and pituitary gland. Using the antibody 305-1A5, all staining seen using MSVA-457R was confirmed except for the staining of specific cells of the epididymis (Appendix A). At the selected conditions, the staining with 305-1A5 also resulted in strong staining of the thyroid (colloid staining), matrix proteins in the aortic wall, and the cytoplasm of intestinal epithelial cells and excretory ducts of salivary glands.

### 3.3. Mammaglobin-A in Neoplastic Tissues

Positive mammaglobin-A immunostaining was detectable in 1450 (10.2%) of the 14,232 analyzable tumors, including 659 (4.6%) with weak, 275 (1.9%) with moderate, and 516 (3.6%) with strong immunostaining. Overall, 37 (28.9%) of 128 tumor categories showed detectable mammaglobin-A staining with 26 (20.3%) tumor categories showing, at least in one case, strong positivity (Table 1). A total of 1437 of 1450 (99%) of all mammaglobin-A-positive tumors were derived from four organs, including the salivary glands (16 tumor categories), breast (6 tumor categories), endometrium (5 tumor categories), and ovary (5 tumor categories). Only five additional tumor types showed occasional mammaglobin-A-positive cases. These tumor categories included medullary thyroid cancer, testicular teratoma, squamous cell carcinoma of the skin, squamous cell carcinoma of the pharynx, and prostatic adenocarcinoma (Gleason 5 + 5 = 10). In these tumors, mammaglobin-A positivity was mostly considered to be weak. Representative images of mammaglobin-A-positive tumors are shown in Figure 2. A graphical representation of a ranking order of mammaglobin-A-positive and strongly positive tumors is given in Figure 3.

### 3.4. Mammaglobin-A Expression, Tumor Phenotype, and Prognosis

Among 1139 evaluable invasive breast carcinomas of NST, low or absent mammaglobin-A immunostaining was linked to a high BRE grade (*p* = 0.0011; Table 2), loss of estrogen receptor and progesterone receptor expression, and triple-negative status (*p* < 0.0001 each) but not to overall survival (Figure 4). Absent or low mammaglobin-A immunostaining was linked to an advanced tumor stage in endometroid endometrium carcinoma (*p* = 0.0198). Although a similar trend was seen for endometroid and serous high-grade carcinomas of the ovary, these associations did not reach statistical significance.

## 4. Discussion

The data of this study identify mammaglobin-A as an oligospecific marker that is expressed in only a few non-vital normal tissues and corresponding tumors.

The International Working Group for Antibody Validation (IWGAV) had proposed that assay validation for immunohistochemistry on formalin fixed tissues should include either a comparison with expression data obtained by another independent method or a comparison with an independent second antibody [32]. Our normal tissue analysis revealed mammaglobin-A immunostaining in only seven organs, including all four organs (breast, uterine cervix, sebaceous glands (skin), and salivary glands) for which RNA expression data have been described in databases resulting from the Human Protein Atlas (HPA) RNA-seq tissue dataset [33], the FANTOM5 project [34,35], and the Genotype-Tissue Expression (GTEx) project [36]. The fact that mammaglobin-A RNA expression was not described for the endometrium, fallopian tube, and epididymis may potentially be due to the small fraction of the total cells of these organs expressing mammaglobin-A, which would result in a marked under-representation of these cells in RNA analyses. The confirmation of positive staining in the endometrium and fallopian tube using the independent mammaglobin-A antibody 305-1A5 identifies these organs as true mammaglobin-A expressors. The positive staining using MSVA-457R in the epididymis were not confirmed using 305-1A5 and may, therefore, reflect a specific cross-reactivity of this antibody. Positive staining in the aortic wall, small intestine, and thyroid were limited to 305-1A5 and may, therefore, constitute specific cross-reactivities of this antibody.

The standardized analysis of 14,232 tumors provided data that were largely reflective of the mammaglobin-A expression pattern in normal tissues. The fact that mammaglobin-A can be expressed in neoplasms derived from the breast [21], ovary [37], uterus [26], and salivary glands [38] was already known. More than 130 studies have analyzed mammaglobin-A expression in these tumors using immunohistochemistry (Figure 5). The diversity of the results of these studies mirrors the usual variability of IHC data that are a logical result of the use of different antibodies, staining protocols, and thresholds for defining “positive” cases [39]. A pivotal result of this study is a ranking order of human tumors according to the prevalence and intensity of mammaglobin-A expression. This enables an assessment of the relative importance of mammaglobin-A across tumor entities. Although the absolute positivity rates described in this study are specific to the reagents and the protocol used in our laboratory, a similar ranking order would be expected if other specific antibodies or different protocols were used. The complete absence of mammaglobin-A immunostaining in several important non-breast and non-gynecological cancer types that share histologic similarities with these tumors, such as adenocarcinomas from the gastrointestinal tract, pancreas, and lung, and cholangiocellular carcinomas, emphasizes the high diagnostic utility of mammaglobin-A immunohistochemistry if the clinical and morphological differential diagnosis includes gynecological tumors. It is also of note that mammaglobin-A immunostaining was not observed in any of 1235 urothelial tumors in this study. This emphasizes the utility of mammaglobin-A in combination with GATA3, a marker that primarily recognizes breast and urothelial neoplasms [40].

At least 11 studies involving 30–1017 patients have earlier analyzed the prognostic relevance of mammaglobin-A expression in breast cancer. Among these, three found a poor prognosis of tumors with high mammaglobin-A expression [20,41,42], five reported a poor prognosis of tumors with low mammaglobin-A expression [25,43,44,45,46], and three did not find a link between mammaglobin expression and patient outcome [47,48,49]. The fact that we were unable to find a significant association between mammaglobin-A expression and patient outcome in 1139 cases of invasive breast cancer of homogeneous histologic subtype (all NST) might suggest that mammaglobin-A expression is not a critical prognostic feature in breast cancer. A complex role of mammaglobin-A in breast cancer is suggested by the documented occurrence of both increased and reduced or even lost mammaglobin-A expression in subsets of breast cancers. A tumor-relevant role for mammaglobin-A upregulation in neoplastic breast epithelium has earlier been proposed by functional studies showing that mammaglobin-A can both promote [16] as well as reduce [17] cell proliferation, migration, and invasion capacities of breast cancer cells. The functional mechanism(s) by which mammaglobin upregulation could influence cancer aggressiveness are not clear. Mammaglobin-A downregulation has been documented in 49% of breast cancers by Zafrakas et al. [14] comparing normal and tumor tissues, and our data shows the complete absence of mammaglobin-A staining in 46% of NST cancers, while normal breast epithelium usually showed detectable staining. A reduced expression of proteins that normally occur in the cells of tumor origin is a sign of tumor cell dedifferentiation, which is often related to unfavorable tumor features [50,51]. This concept may explain associations between reduced mammaglobin-A expression and high tumor grades or unfavorable molecular parameters in breast cancer and an advanced tumor stage in endometrium cancer. In our study, the best patient outcome was seen in breast cancer patients with a moderate mammaglobin-A staining of their tumors, while the prognosis was slightly worse in tumors with weak or negative staining and worst in tumors with strong staining. Based on these—statistically insignificant—data, it could be considered that moderate staining reflects the normal status and that both upregulation and downregulation could be linked to tumor progression.

The fact that mammaglobin-A expression was only found in non-vital tissues in our near complete normal tissue screening involving 76 different tissue categories identifies mammaglobin-A as a potentially attractive therapeutic target. Several studies have indeed evaluated the utility of mammaglobin-A as a drug target and investigated multiple therapeutic approaches. Two studies used adoptive CD8 cytotoxic T-cell transfer and engineered dendritic cells as proof of principle to induce immune responses against mammaglobin-A-positive breast cancer cells in mice [52] and cell cultures [53]. Other studies successfully generated mammaglobin-A-specific CD4 and CD8 T-cell cultures and identified candidate mammaglobin-A epitopes that could serve as antitumor vaccines [54,55,56,57]. One of them led to a clinical phase 1 trial where the safety of a mammaglobin-A DNA vaccine was demonstrated in patients with metastatic breast cancer [58]. At present (last updated January 2023), the authors are recruiting patients for a phase 1b trial (NCT02204098). Recently, novel mammaglobin-A epitopes were reported which could be employed for a specific mammaglobin-A-targeting nanoparticle-conjugated antibody therapy [15].

In summary, our data show that mammaglobin-A can be highly expressed in various tumors derived from the breast, ovary, uterus, and salivary glands. The fact that mammaglobin-A expression was only rarely found in tumors derived from other organs makes mammaglobin-A immunohistochemistry a useful tool to determine the origin of adenocarcinomas, especially in female patients.

## Figures and Tables

**Figure 1 diagnostics-13-01202-f001:**
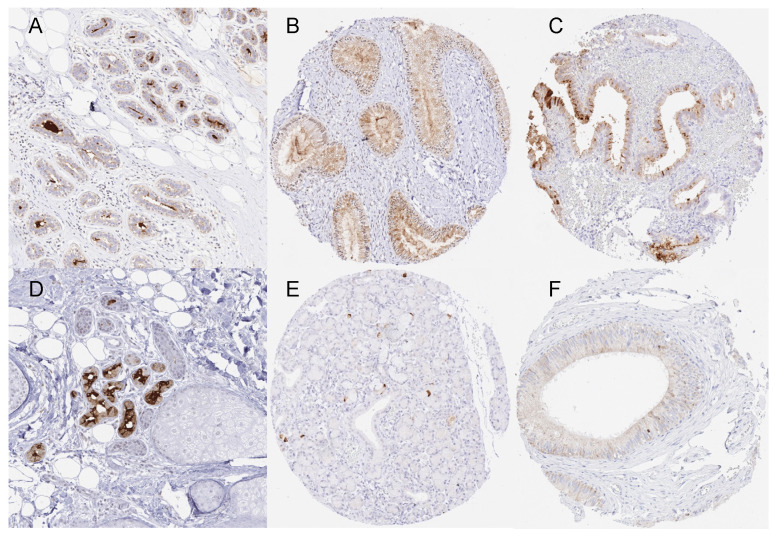
Mammaglobin-A immunostaining of normal tissues. The panels show an apical membranous and cytoplasmic staining of variable intensities in luminal cells of the breast ((**A**), magnification from a TMA spot), endocervical glands (**B**), endometrial glands (**C**), eccrine glands of the skin ((**D**), magnification from a TMA spot), scattered epithelial cells of submandibular gland (**E**), and some chief cells in the corpus epididymis (**F**).

**Figure 2 diagnostics-13-01202-f002:**
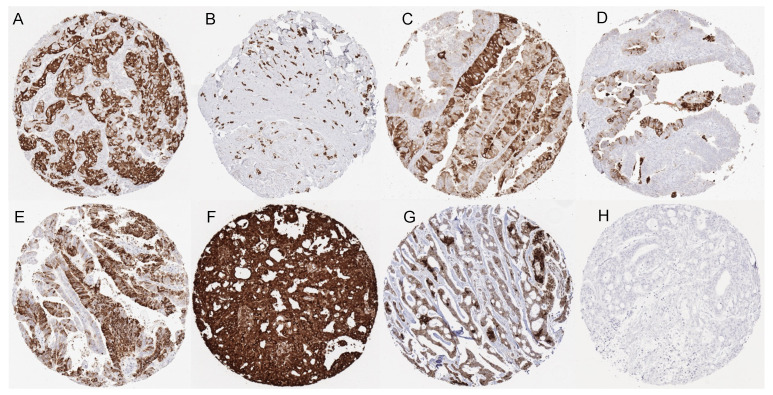
Mammaglobin-A immunostaining in cancer. The panels show distinct, diffuse, or focal mammaglobin-A immunostaining of breast cancer of no special type (**A**), a lobular breast cancer (**B**), an endometrioid (**C**) and a high-grade serous carcinoma (**D**) of the ovary, an endometrioid endometrium carcinoma (**E**) as well as a mucoepidermoid (**F**) and an adenoid cystic (**G**) carcinoma of the salivary gland. Mammaglobin-A staining is absent in the colorectal adenocarcinoma (**H**).

**Figure 3 diagnostics-13-01202-f003:**
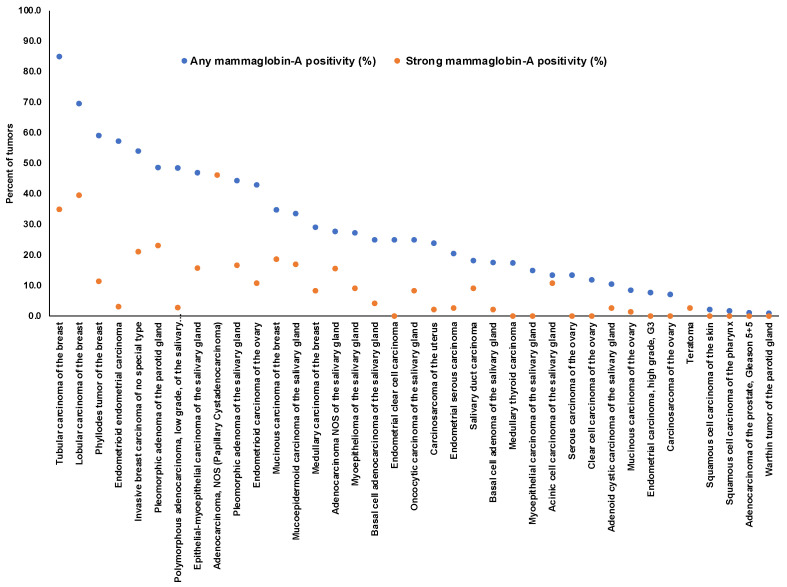
Ranking order of mammaglobin-A immunostaining in tumors. Both the frequency of positive cases (blue dots) and strongly positive cases (orange dots) are shown.

**Figure 4 diagnostics-13-01202-f004:**
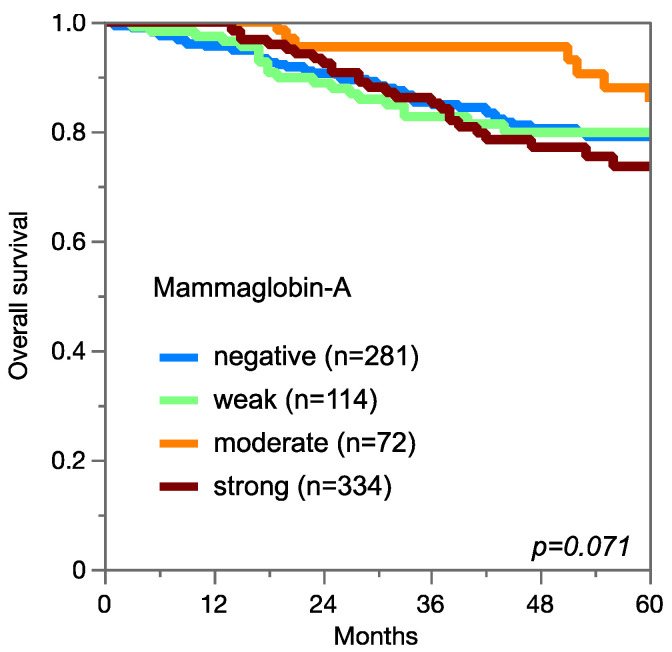
Mammaglobin-A immunostaining and overall survival in breast carcinoma of no special type.

**Figure 5 diagnostics-13-01202-f005:**
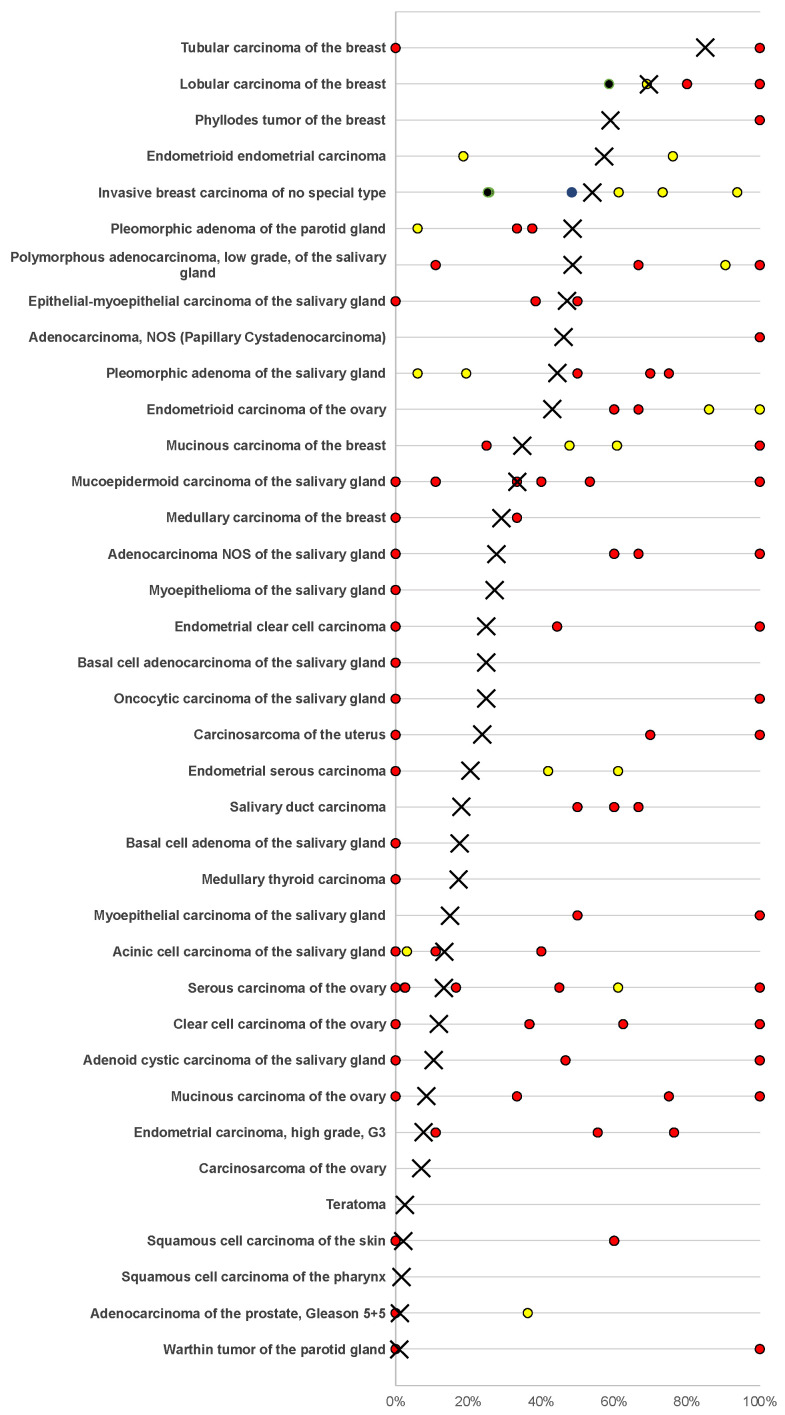
Mammaglobin-A positivity in the literature. An “×” indicates the fraction of mammaglobin-A-positive tumors in the present study. Dots indicate the reported frequencies from the literature for comparison: red dots mark studies with ≤10 tumors, yellow dots mark studies with 11–25 tumors, and black dots mark studies with >25 tumors. All studies are quoted in a list of references in Appendix A.

**Table 1 diagnostics-13-01202-t001:** Mammaglobin-A immunostaining in human tumors.

			Mammaglobin-A Immunostaining				
	Tumor Entity	on TMA (*n*)	Anal. (*n*)	Neg. (%)	Weak (%)	Mod. (%)	Str. (%)
Tumors of the skin	Pilomatricoma	35	32	100.0	0.0	0.0	0.0
	Basal cell carcinoma	88	83	100.0	0.0	0.0	0.0
	Benign nevus	29	29	100.0	0.0	0.0	0.0
	Squamous cell carcinoma of the skin	90	90	97.8	2.2	0.0	0.0
	Malignant melanoma	48	45	100.0	0.0	0.0	0.0
	Merkel cell carcinoma	46	41	100.0	0.0	0.0	0.0
Tumors of the head and neck	Squamous cell carcinoma of the larynx	110	105	100.0	0.0	0.0	0.0
	Squamous cell carcinoma of the pharynx	60	59	98.3	1.7	0.0	0.0
	Oral squamous cell carcinoma (floor of the mouth)	130	129	100.0	0.0	0.0	0.0
	Pleomorphic adenoma of the parotid gland	50	39	51.3	10.3	15.4	23.1
	Warthin tumor of the parotid gland	104	98	99.0	0.0	1.0	0.0
	Adenocarcinoma, NOS (papillary cystadenocarcinoma)	14	13	53.8	0.0	0.0	46.2
	Salivary duct carcinoma	15	11	81.8	9.1	0.0	9.1
	Acinic cell carcinoma of the salivary gland	181	148	86.5	1.4	1.4	10.8
	Adenocarcinoma NOS of the salivary gland	109	83	72.3	6.0	6.0	15.7
	Adenoid cystic carcinoma of the salivary gland	180	114	89.5	7.9	0.0	2.6
	Basal cell adenocarcinoma of the salivary gland	25	24	75.0	20.8	0.0	4.2
	Basal cell adenoma of the salivary gland	101	91	82.4	9.9	5.5	2.2
	Epithelial–myoepithelial carcinoma of the salivary gland	53	51	52.9	23.5	7.8	15.7
	Mucoepidermoid carcinoma of the salivary gland	343	259	66.4	9.7	6.9	17.0
	Myoepithelial carcinoma of the salivary gland	21	20	85.0	5.0	10.0	0.0
	Myoepithelioma of the salivary gland	11	11	72.7	18.2	0.0	9.1
	Oncocytic carcinoma of the salivary gland	12	12	75.0	16.7	0.0	8.3
	Polymorphous adenocarcinoma, low grade, of the salivary gland	41	35	51.4	34.3	11.4	2.9
	Pleomorphic adenoma of the salivary gland	53	36	55.6	13.9	13.9	16.7
Tumors of the lung, pleura, and thymus	Adenocarcinoma of the lung	246	176	100.0	0.0	0.0	0.0
	Squamous cell carcinoma of the lung	130	69	100.0	0.0	0.0	0.0
	Small cell carcinoma of the lung	20	16	100.0	0.0	0.0	0.0
	Mesothelioma, epithelioid	39	28	100.0	0.0	0.0	0.0
	Mesothelioma, biphasic	76	63	100.0	0.0	0.0	0.0
	Thymoma	29	29	100.0	0.0	0.0	0.0
Tumors of the female genital tract	Squamous cell carcinoma of the vagina	78	73	100.0	0.0	0.0	0.0
	Squamous cell carcinoma of the vulva	130	124	100.0	0.0	0.0	0.0
	Squamous cell carcinoma of the cervix	130	125	100.0	0.0	0.0	0.0
	Endometrioid endometrial carcinoma	236	225	42.7	43.6	10.7	3.1
	Endometrial serous carcinoma	82	73	79.5	17.8	0.0	2.7
	Carcinosarcoma of the uterus	48	46	76.1	21.7	0.0	2.2
	Endometrial carcinoma, high grade, G3	13	13	92.3	7.7	0.0	0.0
	Endometrial clear cell carcinoma	8	8	75.0	25.0	0.0	0.0
	Endometrioid carcinoma of the ovary	110	93	57.0	25.8	6.5	10.8
	Serous carcinoma of the ovary, high grade	559	479	86.6	13.2	0.2	0.0
	Mucinous carcinoma of the ovary	96	71	91.5	5.6	1.4	1.4
	Clear cell carcinoma of the ovary	50	42	88.1	11.9	0.0	0.0
	Carcinosarcoma of the ovary	47	42	92.9	4.8	2.4	0.0
	Brenner tumor	9	9	100.0	0.0	0.0	0.0
Tumors of the breast	Invasive breast carcinoma of no special type	1391	1214	45.9	20.6	12.4	21.2
	Lobular carcinoma of the breast	294	260	30.4	21.2	8.8	39.6
	Medullary carcinoma of the breast	26	24	70.8	12.5	8.3	8.3
	Tubular carcinoma of the breast	27	20	15.0	35.0	15.0	35.0
	Mucinous carcinoma of the breast	58	43	65.1	9.3	7.0	18.6
	Phyllodes tumor of the breast	50	44	40.9	34.1	13.6	11.4
Tumors of the digestive system	Adenomatous polyp, low-grade dysplasia	50	50	100.0	0.0	0.0	0.0
	Adenomatous polyp, high-grade dysplasia	50	49	100.0	0.0	0.0	0.0
	Adenocarcinoma of the colon	1932	1808	100.0	0.0	0.0	0.0
	Gastric adenocarcinoma, diffuse type	226	161	100.0	0.0	0.0	0.0
	Gastric adenocarcinoma, intestinal type	224	167	100.0	0.0	0.0	0.0
	Gastric adenocarcinoma, mixed type	62	56	100.0	0.0	0.0	0.0
	Adenocarcinoma of the esophagus	133	82	100.0	0.0	0.0	0.0
	Squamous cell carcinoma of the esophagus	124	71	100.0	0.0	0.0	0.0
	Squamous cell carcinoma of the anal canal	91	85	100.0	0.0	0.0	0.0
	Cholangiocarcinoma	50	49	100.0	0.0	0.0	0.0
	Hepatocellular carcinoma	50	50	100.0	0.0	0.0	0.0
	Ductal adenocarcinoma of the pancreas	662	593	100.0	0.0	0.0	0.0
	Pancreatic/ampullary adenocarcinoma	119	86	100.0	0.0	0.0	0.0
	Acinar cell carcinoma of the pancreas	14	13	100.0	0.0	0.0	0.0
	Gastrointestinal stromal tumor (GIST)	50	49	100.0	0.0	0.0	0.0
Tumors of the urinary system	Non-invasive papillary urothelial carcinoma, pTa G2 low grade	177	141	100.0	0.0	0.0	0.0
	Non-invasive papillary urothelial carcinoma, pTa G2 high grade	141	117	100.0	0.0	0.0	0.0
	Non-invasive papillary urothelial carcinoma, pTa G3	187	113	100.0	0.0	0.0	0.0
	Urothelial carcinoma, pT2-4 G3	1207	825	100.0	0.0	0.0	0.0
	Small cell neuroendocrine carcinoma of the bladder	18	18	100.0	0.0	0.0	0.0
	Sarcomatoid urothelial carcinoma	25	21	100.0	0.0	0.0	0.0
	Clear cell renal cell carcinoma	858	824	100.0	0.0	0.0	0.0
	Papillary renal cell carcinoma	255	232	100.0	0.0	0.0	0.0
	Clear cell (tubulo) papillary renal cell carcinoma	21	20	100.0	0.0	0.0	0.0
	Chromophobe renal cell carcinoma	131	122	100.0	0.0	0.0	0.0
	Oncocytoma	177	162	100.0	0.0	0.0	0.0
Tumors of the male genital organs	Adenocarcinoma of the prostate, Gleason 3 + 3	83	83	100.0	0.0	0.0	0.0
	Adenocarcinoma of the prostate, Gleason 4 + 4	80	80	100.0	0.0	0.0	0.0
	Adenocarcinoma of the prostate, Gleason 5 + 5	85	85	98.8	0.0	1.2	0.0
	Adenocarcinoma of the prostate (recurrence)	261	254	100.0	0.0	0.0	0.0
	Small cell neuroendocrine carcinoma of the prostate	17	15	100.0	0.0	0.0	0.0
	Seminoma	621	611	100.0	0.0	0.0	0.0
	Embryonal carcinoma of the testis	50	44	100.0	0.0	0.0	0.0
	Yolk sac tumor	50	43	100.0	0.0	0.0	0.0
	Teratoma	50	38	97.4	0.0	0.0	2.6
	Squamous cell carcinoma of the penis	80	79	100.0	0.0	0.0	0.0
Tumors of endocrine organs	Adenoma of the thyroid gland	50	47	100.0	0.0	0.0	0.0
	Papillary thyroid carcinoma	50	48	100.0	0.0	0.0	0.0
	Follicular thyroid carcinoma	49	49	100.0	0.0	0.0	0.0
	Medullary thyroid carcinoma	50	46	82.6	13.0	4.3	0.0
	Anaplastic thyroid carcinoma	26	24	100.0	0.0	0.0	0.0
	Adrenal cortical adenoma	50	44	100.0	0.0	0.0	0.0
	Adrenal cortical carcinoma	26	25	100.0	0.0	0.0	0.0
	Phaeochromocytoma	50	48	100.0	0.0	0.0	0.0
	Pancreas, neuroendocrine tumor (NET)	46	42	100.0	0.0	0.0	0.0
	Pancreas, neuroendocrine carcinoma (NEC)	3	3	100.0	0.0	0.0	0.0
Tumors of hematopoietic and lymphoid tissues	Hodgkin lymphoma	103	100	100.0	0.0	0.0	0.0
	Non-Hodgkin lymphoma	62	61	100.0	0.0	0.0	0.0
	Small lymphocytic lymphoma, B-cell type (B-SLL/B-CLL)	50	50	100.0	0.0	0.0	0.0
	Diffuse large B-cell lymphoma (DLBCL)	114	114	100.0	0.0	0.0	0.0
	Follicular lymphoma	88	88	100.0	0.0	0.0	0.0
	T-cell non-Hodgkin lymphoma	24	24	100.0	0.0	0.0	0.0
	Mantle cell lymphoma	18	18	100.0	0.0	0.0	0.0
	Marginal zone lymphoma	16	16	100.0	0.0	0.0	0.0
	Diffuse large B-cell lymphoma (DLBCL) in the testis	16	16	100.0	0.0	0.0	0.0
	Burkitt lymphoma	5	3	100.0	0.0	0.0	0.0
Tumors of soft tissue and bone	Tendosynovial giant cell tumor	45	43	100.0	0.0	0.0	0.0
	Granular cell tumor	53	43	100.0	0.0	0.0	0.0
	Leiomyoma	50	47	100.0	0.0	0.0	0.0
	Leiomyosarcoma	87	87	100.0	0.0	0.0	0.0
	Liposarcoma	132	121	100.0	0.0	0.0	0.0
	Malignant peripheral nerve sheath tumor (MPNST)	13	11	100.0	0.0	0.0	0.0
	Myofibrosarcoma	26	26	100.0	0.0	0.0	0.0
	Angiosarcoma	73	67	100.0	0.0	0.0	0.0
	Angiomyolipoma	91	88	100.0	0.0	0.0	0.0
	Dermatofibrosarcoma protuberans	21	18	100.0	0.0	0.0	0.0
	Ganglioneuroma	14	14	100.0	0.0	0.0	0.0
	Kaposi sarcoma	8	5	100.0	0.0	0.0	0.0
	Neurofibroma	117	116	100.0	0.0	0.0	0.0
	Sarcoma, not otherwise specified (NOS)	75	71	100.0	0.0	0.0	0.0
	Paraganglioma	41	41	100.0	0.0	0.0	0.0
	Primitive neuroectodermal tumor (PNET)	23	16	100.0	0.0	0.0	0.0
	Rhabdomyosarcoma	7	7	100.0	0.0	0.0	0.0
	Schwannoma	121	118	100.0	0.0	0.0	0.0
	Synovial sarcoma	12	11	100.0	0.0	0.0	0.0
	Osteosarcoma	43	36	100.0	0.0	0.0	0.0
	Chondrosarcoma	38	17	100.0	0.0	0.0	0.0

**Table 2 diagnostics-13-01202-t002:** Mammaglobin-A immunostaining and tumor phenotypes in breast cancers of no special type, endometroid endometrium carcinoma, as well as endometroid and serous high-grade ovarian cancers.

				Mammaglobin-A IHC				
			*n*	Negative (%)	Weak (%)	Moderate (%)	Strong (%)	*p*
Breast cancer of no special type (NST)	Tumor stage	pT1	601	44.3	21.5	14.1	20.1	0.1421
		pT2	411	46.5	18.5	11.7	23.4	
		pT3–4	84	54.8	16.7	6.0	22.6	
	Grade	G1	176	37.5	21.6	13.6	27.3	0.0011
		G2	590	41.9	21.7	13.7	22.7	
		G3	372	54.3	19.4	10.8	15.6	
	Nodal stage	pN0	515	44.1	21.6	14.6	19.8	0.3017
		pN1	223	44.8	19.7	13.0	22.4	
		pN2	69	53.6	15.9	5.8	24.6	
		pN3	56	50.0	14.3	8.9	26.8	
	HER2 status	neg	851	47.1	20.8	11.2	20.9	0.4501
		pos	118	43.2	22.0	16.1	18.6	
	ER status	neg	200	63.5	14.5	9.5	12.5	<0.0001
		pos	717	41.6	23.0	12.8	22.6	
	PR status	neg	390	55.4	17.7	10.5	16.4	<0.0001
		pos	568	40.0	23.2	13.2	23.6	
	Triple negative	no	756	41.7	22.6	13.4	22.4	<0.0001
		yes	134	75.4	10.4	5.2	9.0	
Endometrioid endometrial cancers	Tumor stage	pT1	113	33.6	48.7	14.2	3.5	0.0198
		pT2	24	50.0	37.5	12.5	0.0	
		pT3–4	35	48.6	51.4	0.0	0.0	
		pN0	50	36.0	56.0	6.0	2.0	0.0986
	Nodal stage	pN+	30	63.3	33.3	3.3	0.0	
Endometrioid ovarian cancer	Tumor stage	pT1	22	40.9	31.8	9.1	18.2	0.1686
		pT2	4	100.0	0.0	0.0	0.0	
		pT3	5	80.0	20.0	0.0	0.0	
	Nodal stage	pN0	20	40.0	35.0	10.0	15.0	0.3835
		pN1	7	71.4	14.3	0.0	14.3	
Serous ovarian cancers	Tumor stage	pT1	31	77.4	22.6	0.0	0.0	0.2
		pT2	40	92.5	7.5	0.0	0.0	
		pT3	245	88.2	11.8	0.0	0.0	
	Nodal stage	pN0	78	83.3	16.7	0.0	0.0	0.3767
		pN1	153	87.6	12.4	0.0	0.0	

## Data Availability

All data generated or analyzed during this study are included in this published article.

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
