# Peer review of "Mammaglobin-A Expression Is Highly Specific for Tumors Derived from the Breast, the Female Genital Tract, and the Salivary Gland"

_diagnostics, 2023, doi:10.3390/diagnostics13061202_

Round 1
Reviewer 1 Report
This a well-writen study and despite several previous studies do exist on the subject, it is desirbale to have more well designed studies with such a large number of specimens and tumor types. Few suggestions:
1/ Regarding TMAs: one spot/sample? Usually 3 spots from the same sample. To define in the text.
2/ The 2 antibodies were used for both TMAs? Why in the results the second antibody is not mentioned? Only one used for these TMAs? To clarify.
3/ The first antibody was used in manual technique? To clarify.
4/ I understand that according to MDPI template, the Table should appear like this, but right now it is impossible to read. The Editorial Office should find a better way to present it (different layout, lines?).
5/ Advanced stage for all endometrioid (no grade?)?
6/ To add high grade serous carcinome for the ovary (Table), if all cases were high grade.
7/ Figure 5 is indeed useful for the readers. But, how the authors created it? References of these studies should at least be mentioned (even if it is in sup files)
Author Response
Reviewer 1
This a well-writen study and despite several previous studies do exist on the subject, it is desirbale to have more well designed studies with such a large number of specimens and tumor types. Few suggestions:
1/ Regarding TMAs: one spot/sample? Usually 3 spots from the same sample. To define in the text.
Reply: We have now clarified that one 0.6 mm tissue core per tumor was used for TMA making (Materials & Methods section “Tissue Microarrays”.
2/ The 2 antibodies were used for both TMAs? Why in the results the second antibody is not mentioned? Only one used for these TMAs? To clarify.
Reply: We have now clearly defined which antibodies were used for normal tissue and tumor tissue analysis in the Materials & Methods section “Immunohistochemistry”. The results of the second antibody are given in the “Results” section (3.2. Mammaglobin-A in normal tissue, page 4, line 138 and following).
3/ The first antibody was used in manual technique? To clarify.
Reply: We have now clarified that the first antibody was stained manually (Materials & Methods section “Immunohistochemistry”).
4/ I understand that according to MDPI template, the Table should appear like this, but right now it is impossible to read. The Editorial Office should find a better way to present it (different layout, lines?).
Reply: We adjusted the column width to increase readability.
5/ Advanced stage for all endometrioid (no grade?)?
Reply: Unfortunately, we do not have grade information available.
6/ To add high grade serous carcinome for the ovary (Table), if all cases were high grade.
Reply: We have now specified in table 1 that these were all high-grade carcinomas.
7/ Figure 5 is indeed useful for the readers. But, how the authors created it? References of these studies should at least be mentioned (even if it is in sup files)
Reply: We have compiled the literature on figure 5 in the supplementary table 1.
Reviewer 2 Report
The authors analyzed immunolocalization of mammaglobin A in both normal and neoplastic tissues of human and revealed that the expression of mammaglobin A is highly specific in for tumors derived from the breast, the female genital tract and the salivary gland.
The conclusion might reliable because this study covered many kinds of tissues with large sample size.
Please address following points.
1. Results from normal tissue immunohistochemistry should be described in the abstract.
2. Are paired samples (tumor/normal from same patients) included in the study? If so, it is interesting to compare the immunostaining of mammaglobin A in paired samples.
3. How about evaluating immunostaings based on only the intensity? Evaluating area is limited in TMA and it seems difficult to exactly evaluate staining area.
4. It is interesting to analyze prognostic value of mammaglobin A in breast cancers according to subtypes of breast cancers (for example, TNBC or not).
Minor
1. Are the pictures of Figure 1A and 1D really from TMA tissues?
2. Figure legend of Figure 1 should be carefully checked. The pictures are A-F, while the legend is A, B, C, E, F, G.
Author Response
Reviewer 2
The authors analyzed immunolocalization of mammaglobin A in both normal and neoplastic tissues of human and revealed that the expression of mammaglobin A is highly specific in for tumors derived from the breast, the female genital tract and the salivary gland.
The conclusion might reliable because this study covered many kinds of tissues with large sample size.
Please address following points.
- Results from normal tissue immunohistochemistry should be described in the abstract.
Reply: We have added normal tissue results to the abstract.
- Are paired samples (tumor/normal from same patients) included in the study? If so, it is interesting to compare the immunostaining of mammaglobin A in paired samples.
Reply: There were no matched normal tissues / tumor tissues in our study.
- How about evaluating immunostainings based on only the intensity? Evaluating area is limited in TMA and it seems difficult to exactly evaluate staining area.
Reply: We use a standard scoring system based on a combination of staining intensity and fraction of stained tumor cells which is designed to minimize the impact of staining intensity variations. We added a reference to our scoring system in the Materials and Methods section (“Immunohistochemistry”).
- It is interesting to analyze prognostic value of mammaglobin A in breast cancers according to subtypes of breast cancers (for example, TNBC or not).
Reply: We agree that subset analyses would be of interest, but, unfortunately, the number of (mammaglobin-A positive) cases in the subsets of interest (e.g. TNBC) is too small for meaningful statistical analysis.
Minor
- Are the pictures of Figure 1A and 1D really from TMA tissues?
Reply: We have added the information that 1A and 1D show magnifications from a TMA spot to the legend to Figure 1.
- Figure legend of Figure 1 should be carefully checked. The pictures are A-F, while the legend is A, B, C, E, F, G.
Reply: The reviewer is right. This was a typo, and we corrected the legend accordingly.

Round 2
Reviewer 2 Report
The manuscript is well revised.